# CANDIDATE LABEL SET PRUNING: A DATA-CENTRIC PERSPECTIVE FOR DEEP PARTIAL-LABEL LEARNING

**Shuo He[1]   Chaojie Wang[2]   Guowu Yang[1]\*   Lei Feng[2]\***

[1]University of Electronic Science and Technology of China
[2]Nanyang Technological University
shuohe123@gmail.com, chaojie.wang@ntu.edu.sg
guowu@uestc.edu.cn, lfengqaq@gmail.com

## ABSTRACT

Partial-label learning (PLL) allows each training example to be equipped with a set of candidate labels where only one is the true label. Existing deep PLL research focuses on a *learning-centric* perspective to design various training strategies for label disambiguation i.e., identifying the concealed true label from the candidate label set for model training. However, when the size of the candidate label set becomes excessively large, these learning-centric strategies would be unable to find the true label for model training, thereby causing performance degradation. This motivates us to think from a *data-centric* perspective and pioneer a new PLL-related task called *candidate label set pruning* (CLSP) that aims to filter out certain potential false candidate labels in a training-free manner. To this end, we propose the first CLSP method based on the inconsistency between the representation space and the candidate label space. Specifically, for each candidate label of a training instance, if it is not a candidate label of the instance's nearest neighbors in the representation space, then it has a high probability of being a false label. Based on this intuition, we employ a per-example pruning scheme that filters out a specific proportion of high-probability false candidate labels. Theoretically, we prove an upper bound of the pruning error rate and analyze how the quality of representations affects our proposed method. Empirically, extensive experiments on both benchmark-simulated and real-world PLL datasets validate the great value of CLSP to significantly improve many state-of-the-art deep PLL methods.

## 1 INTRODUCTION

The effective training of modern deep neural networks (DNNs) commonly requires a large amount of perfectly labeled data, which imposes a great challenge on data annotations. However, better data quality is normally at odds with lower labeling costs in many real-world applications (Zhou, 2018). To achieve a rational trade-off, partial-label learning (PLL), a coarse-grained labeling scheme that allows assigning a candidate label set for each training example (Cour et al., 2011), has attracted increasing attention in recent years (Wang et al., 2022b; Xu et al., 2023b). PLL naturally arises in many real-world applications, such as web news annotation (Luo & Orabona, 2010), automatic image annotation (Chen et al., 2017), and multi-media analysis (Zeng et al., 2013).

Existing deep PLL research focuses on a learning-centric perspective to design various training strategies for label disambiguation, i.e., identifying the true label from the candidate label set for model training, e.g., progressive purification (Lv et al., 2020; Wu et al., 2022), class activation value (Zhang et al., 2022), class prototypes (Wang et al., 2022b). However, the excessively large magnitude of the candidate label set inevitably brings an obstacle to label disambiguation (i.e., identifying the concealed true label) for these learning-centric deep PLL methods, because the misidentified labels could have profound negative impacts on model training throughout the training process.

In this paper, we pioneer a new PLL-related task called *candidate label set pruning* (CLSP). CLSP aims to filter out potential false candidate labels in a training-free manner, instead of learning DNNs

---

\*Corresponding authors.

with various training strategies for label disambiguation in conventional deep PLL research. To this end, we propose the first versatile training-free CLSP method, based on the *inconsistency* between the representation space and the candidate label space. Specifically, for each candidate label of a training instance, if it is not a candidate label of the instance's nearest neighbors in the representation space, then it has a high probability of being a false label. Based on this intuition, we employ a per-example pruning scheme that filters out a specific proportion of high-probability false candidate labels. Theoretically, we prove an upper bound of the pruning error rate and analyze how the quality of representations affects the proposed algorithm. Empirically, we evaluate the task of CLSP on both benchmark-simulated and real-world datasets across various PLL settings with eleven state-of-the-art deep PLL methods. Extensive experiments clearly validate the effectiveness of our proposed CLSP method to improve existing PLL methods.

Our main contributions can be summarized as follows:

- *A new data-centric perspective for deep PLL.* Different from the conventional learning-centric perspective in deep PLL research, we pioneer a new PLL-related task called candidate label set pruning (CLSP) to improve existing deep PLL methods.

- *A versatile efficient algorithm.* We propose the first versatile training-free CLSP algorithm that prunes a certain proportion of candidates based on the inconsistency between the representation space and candidate label space.

- *Theoretical analysis.* We theoretically prove an upper bound of the per-example pruning error rate and analyze how the representation quality affects the proposed algorithm.

- *Significant experimental improvements.* We perform comprehensive experiments on four benchmarks under various PLL settings with eleven state-of-the-art deep PLL methods. Significant improvement validates the superiority of the proposed CLSP method.

## 2  RELATED WORKS

**Conventional partial-label learning.** Early exploration of PLL before the trend of deep learning techniques focused on small-scale datasets with hand-crafted features (Gong et al., 2022). There are mainly two different strategies to handle candidate labels: *averaging* and *identification*. The former treats all candidate labels equally (Cour et al., 2011), while the latter aims to identify the concealed true label from candidate labels (Zhang et al., 2016; Xu et al., 2019; Lyu et al., 2019). The drawback of this line of work lies in its limited ability to scale to modern large datasets due to its heavy reliance on hand-crafted features, native linear models, and cost-prohibitive optimization algorithms.

**Deep partial-label learning.** To address the above limitations, recent research of PLL has been focusing on end-to-end training of DNNs (Yan & Guo, 2023b; He et al., 2022). Early deep PLL works studied the consistency of the classification risk and the learned classifier (Feng et al., 2020; Lv et al., 2020; Wen et al., 2021). Due to the unsatisfying practical performance of these methods, subsequent works focused on designing various powerful training strategies for effective label disambiguation, based on class activation value (Zhang et al., 2022), class prototypes (Wang et al., 2022b), or consistency regularization (Wu et al., 2022). Moreover, recent works started to consider more realistic PLL scenarios, including label-dependent (Wang et al., 2022b), instance-dependent (Xu et al., 2021; Xia et al., 2022; Qiao et al., 2023; He et al., 2023b), and noisy (Yan & Guo, 2023a; He et al., 2023a) candidate label set generation, and long-tailed data distribution (Wang et al., 2022a; Hong et al., 2023). In particular, recent work (Xu et al., 2023b) proposed a theoretically grounded algorithm to filter out false candidate labels progressively in every epoch. In summary, existing works in both conventional and deep PLL focus on a learning-centric perspective and inevitably suffer from performance degradation when the size of candidate label sets is excessively large.

**Dataset pruning and distillation.** Dataset pruning (distillation) aims to select (synthesize) a small but representative subset from the entire training data as a proxy set without significantly sacrificing the model's performance. Existing methods generally achieved this goal by certain sample-related characteristics, e.g., diversity (Sener & Savarese, 2018), forgetfulness (Toneva et al., 2018), gradient norm (Paul et al., 2021), training trajectories (Zhao et al., 2021), and the gradients and features (Zhao et al., 2021). These methods focused on reducing the number of training data and meanwhile maintaining the data utility, while our studied CLSP aims to reduce the magnitude of candidate label sets and facilitate existing deep PLL methods.

$k$-**NN for partial-label learning.** The $k$ nearest neighbors ($k$-NN) algorithm is a well-known non-parametric classification technique, which also has been applied to PLL (Hüllermeier & Beringer, 2006) by aggregating the candidate labels of nearest neighbors to predict the label for each instance. Besides, some conventional PLL methods (Zhang & Yu, 2015; Zhang et al., 2016; Xu et al., 2019) also employed $k$-NN-based topological structure in the feature space to refine candidate labels and thus identified the true label for model training. However, these methods focus on learning-centric solutions and still suffer from the excessively large magnitude of candidate label sets.

## 3   A DATA-CENTRIC APPROACH FOR CANDIDATE LABEL SET PRUNING

In this section, we first formally define the task of CLSP, then introduce the proposed method for CLSP, and finally present corresponding theoretical analyses of the pruning error.

### 3.1   PROBLEM SETUP

We start by introducing some necessary symbols and terminologies to define the task of PLL and CLSP formally. Let $\mathcal{X} \in \mathbb{R}^d$ be the $d$-dimensional feature space and $\mathcal{Y} = \{1, 2, ..., c\}$ be the label space with $c$ class labels, respectively. In PLL, each instance $\boldsymbol{x}_i \in \mathcal{X}$ is associated with a candidate label set $Y_i \in 2^{\mathcal{Y}} \backslash \{\varnothing, \mathcal{Y}\}$ that consists of a true label $y_i \in \mathcal{Y}$ and a false candidate label set $Y_i' = Y_i \backslash \{y_i\}$ containing $(|Y_i| - 1)$ redundant false candidate labels $y_i' \in \mathcal{Y}$. In particular, the PLL assumption is that the true label $y_i$ must be in the candidate label set $Y_i$. Given a PLL dataset $\mathcal{D} = \{\boldsymbol{x}_i, Y_i\}_{i=1}^n$, the objective of deep PLL is to learn a deep neural network, which can predict the label of test data as accurately as possible. In contrast, CLSP aims to reduce the size of the candidate label set towards each training example so that the pruned candidate label set can be leveraged by existing deep PLL methods to achieve better performance. Below we introduce two important metrics to evaluate the pruning performance of any pruning method.

**Definition 1** ($\alpha$-error and $\beta$-coverage pruning). *Given a PLL dataset $\mathcal{D} = \{\boldsymbol{x}_i, Y_i\}_{i=1}^n$, for each candidate label set $Y_i$, let $\widetilde{Y}_i$ denote the set of eliminated candidate labels from $Y_i$ and $\bar{Y}_i$ denote the pruned candidate label set of $Y_i$ (i.e., $\bar{Y}_i = Y_i \backslash \widetilde{Y}_i$). The pruning method is $\alpha$-error where $\alpha = \frac{\sum_{i=1}^n \mathbb{I}[y_i \in \widetilde{Y}_i]}{n}$ and $\beta$-coverage where $\beta = \frac{\sum_{i=1}^n |\widetilde{Y}_i|}{\sum_{i=1}^n (|Y_i| - 1)}$.*

Ideally, the optimal pruning of CLSP is to achieve $\alpha = 0$ and $\beta = 1$, which can perfectly find out all false candidate labels and thus finally identify the true label.

### 3.2   THE PROPOSED PRUNING METHOD

Here, we introduce the proposed CLSP method that aims to eliminate potential false candidate labels in the candidate label set of each training PLL instance. Differently from conventional learning-centric deep PLL methods that focus on training a desired deep neural network, we pursue a data-centric solution that delves into the PLL data itself. Motivated by the clusterability setting in the conventional supervised learning (Zhu et al., 2022), we further focus on a label distinguishability setting on the PLL dataset, i.e., nearby PLL instances are likely to have the same true label (vs. false candidate label) inside their candidate label sets with a high probability (vs. a limited probability), which could be formally defined below.

**Definition 2** (($k, \delta_k, \rho_k$)-label distinguishability). *A PLL dataset $\mathcal{D} = \{(\boldsymbol{x}_i, Y_i)\}_{i=1}^n$ satisfies $(k, \delta_k, \rho_k)$ label distinguishability if: $\forall (\boldsymbol{x}_i, Y_i) \in \mathcal{D}$, the true label $y_i \in Y_i$ is inside the candidate label set $Y_i^{(j)}$ of its each $k$-NN example $(\boldsymbol{x}_i^{(j)}, Y_i^{(j)})$, with probability at least $1 - \delta_k$, and each false candidate label $y_i' \in Y_i \backslash \{y_i\}$ is inside the candidate label set $Y_i^{(j)}$ of its each $k$-NN example $(\boldsymbol{x}_i^{(j)}, Y_i^{(j)})$ with probability no more than $\rho_k$.*

Definition 2 characterizes the candidate label distribution in the local representation space, which has two-fold implications. First, the true label of instances in the local representation space appears in their candidate label sets with a high probability. Second, each false candidate label of instances in the local representation space appears in their candidate label sets with a limited probability.

Intuitively, the candidate label of an instance that appears most frequently in its $k$-NN instances' candidate label sets is more likely to be the true label, and the candidate label that rarely appears in

its $k$-NN instances' candidate label sets has a high probability of being a false label. Motivated by this intuition, we introduce a per-instance label-wise metric $O_{ij}$ towards the $i$-th instance $(\boldsymbol{x}_i, Y_i)$ associated with the $j$-th candidate label, to measure the possibility of the $j$-th candidate label is *not* the true label of the instance $\boldsymbol{x}_i$, which implies that we should prune the $j$-th label from the candidate label set if $O_{ij}$ is large. Concretely, $O_{ij}$ is calculated by counting the times of its $j$-th candidate label not appearing in its $k$-NN instances' candidate label set $Y_i^{(k)}$:

$$O_{ij} = \sum\nolimits_{v=1}^{k} \mathbb{I}[y_{ij} \neq y_{ij}^{(v)}], \forall j \in Y_i, \tag{1}$$

where $\mathbb{I}[\cdot]$ is the indicator function and $y_{ij}$ is the $j$-th candidate label of $\boldsymbol{x}_i$. In this way, the value of $O_{ij}$ denotes the number of the $k$-NN instances of $\boldsymbol{x}_i$ whose candidate label set does not include the $j$-th candidate label of $\boldsymbol{x}_i$. Particularly, the calculating process in Eq. (1) is like a "down-voting" procedure where each nearest neighbor instance of the concerned instance serves a voter to down-vote the candidate labels that are not inside its candidate label set $Y_i^{(k)}$. Furthermore, we define a specified parameter $\tau$ to control the pruning extent. Specifically, the number of eliminated candidate labels of $\boldsymbol{x}_i$ is

$$\gamma_i = \lceil \tau(|Y_i| - 1) \rceil, \tag{2}$$

where $\lceil \cdot \rceil$ is the ceiling function that returns the least integer greater than or equal to the given number. Then, for each instance $(\boldsymbol{x}_i, Y_i)$, we can eliminate a subset of candidate labels that possess a high down-voting value:

$$\widetilde{Y}_i = \text{Top-}\gamma_i\text{-argmax}_{j \in Y_i}(O_{ij}), \tag{3}$$

where Top-$\gamma_i$-argmax returns a subset of indices (i.e., candidate labels) that have the highest $\gamma_i$ down-voting values of $O_{ij}$ for $j \in Y_i$. After eliminating the selected subset of candidate labels $\widetilde{Y}_i$ from $Y_i$ for each instance $\boldsymbol{x}_i$, we can obtain the pruned PLL dataset $\bar{\mathcal{D}} = \{(\boldsymbol{x}_i, \bar{Y}_i)\}_{i=1}^{n}$ where $\bar{Y}_i = Y_i \backslash \widetilde{Y}_i$. The pseudo-code of the proposed algorithm for CLSP is shown in Appendix A. Notably, an incorrectly pruned PLL instance $(\boldsymbol{x}_i, Y_i)$ whose true label $y_i$ is inside the eliminated candidate label set $\widetilde{Y}_i$ would become a noisy PLL instance (Yan & Guo, 2023a; Xu et al., 2023a; Lv et al., 2023; Wang et al., 2024), which is more challenging for conventional PLL methods. To alleviate this issue, we would like to analyze the upper bound of the pruning error in the proposed method.

### 3.3 THEORETICAL ANALYSIS OF PRUNING ERROR

Given an example $(\boldsymbol{x}_i, Y_i)$ and its $k$-NN examples $\{(\boldsymbol{x}_i^{(j)}, Y_i^{(j)})\}_{j=1}^{k}$ in a PLL dataset $\mathcal{D}$ that satisfies the $(k, \delta_k, \rho_k)$ label distinguishability, the probability of the true label $y_i \in Y_i$ appearing in each $k$-NN instance's candidate label set $Y_i^{(j)}$ is denoted by $t \geqslant 1 - \delta_k$, and correspondingly the probability of each false candidate label $y_i' \in Y_i' = Y_i \backslash \{y_i\}$ appearing in each $k$-NN instance's candidate label set $Y_i^{(j)}$ is denoted by $q \leqslant \rho_k$. We assume that the true label and false candidate labels of each PLL example appear in its $k$-NN examples' candidate label sets independently. Then, the down-voting statistic of the true label $O_{iy}$ follows a binomial distribution $B(k, 1-t)$, and the down-voting statistic of each false candidate label $O_{iy'}$ follows a binomial distribution $B(k, 1 - q)$. In this case, there are one random variable $O_{iy} \sim B(k, 1 - t)$ and $|Y_i'|$ *i.i.d.* random variables $O_{iy'} \sim B(k, 1 - q)$. Given the number of eliminated candidate labels $\gamma_i \in [1, |Y_i'|]$, an incorrect pruning event for a PLL example $(\boldsymbol{x}_i, Y_i)$ occurs in the proposed algorithm when $O_{iy} \in \text{Top-}\gamma_i\text{-}\arg\max_j(O_{ij})$. In other words, the incorrect pruning implies that the down-voting statistic of the true label $O_{iy}$ is larger than the $\gamma_i$-th highest down-voting statistic of the false candidate label. Formally, based on the definition of the $k$-th *order statistic* $O_{iy'}^{(k)}$ which is the $k$-th item in the ascending order statistics $[O_{iy'}^{(1)}, O_{iy'}^{(k)}, \cdots, O_{iy'}^{(|Y_i'|)}]$, the incorrect pruning event can be defined as $[O_{iy'}^{(|Y_i'| - \gamma_i + 1)} < O_{iy}]$. An upper bound for the probability of getting such an event is shown below.

**Theorem 1.** *Assume that the $(k, \delta_k, \rho_k)$-label distinguishability is satisfied. For each PLL example $(\boldsymbol{x}_i, Y_i)$, let us denote that the $y$-th label in the candidate label set $Y_i$ is the true label, and the $y'$-th label in the false candidate label set $Y_i' = Y_i \backslash \{y\}$ is an arbitrary false candidate label, i.e., $y' \neq y$. Given the number of eliminated candidate labels $\gamma_i$, then the probability of getting an incorrect pruning can be upper bounded by*

$$\mathbb{P}(O_{iy'}^{(\xi_i)} < O_{iy}) \leqslant \sum\nolimits_{j=1}^{k} \sum\nolimits_{m=\xi_i}^{|Y_i'|} \binom{|Y_i'|}{m} \eta^m (1 - \eta)^{(|Y_i'| - m)} b_{\delta_k}(k, j), \tag{4}$$

Figure 1: Numerical simulation experiments to show the effect of $k$ and $\gamma_i$ for the upper bound.

*where $\xi_i = (|Y_i'| - \gamma_i + 1)$, $\binom{n}{r} = \frac{n!}{r!(n-r)!}$ is the combination formula, $b_{\delta_k}(k, j) = \binom{k}{j}\delta_k^j(1-\delta_k)^{k-j}$ denotes the probability mass function of a binomial distribution $B(k, \delta_k)$, and $\eta = I_{\rho_k}(k-j+1, j)$ where $I_{\rho_k}(k, j) = \int_0^{\rho_k} t^{k-1}(1-t)^{j-1}dt$ is the regularized incomplete beta function.*

The proof is provided in Appendix B. Note that the above upper bound is too complicated to perform the mathematical quantitative analysis for four key factors $k$, $\gamma_i$, $\delta_k$, and $\rho_k$. Hence, we aim to conduct an empirical analysis of the four factors to derive technical insights. Specifically, in the practical scenario of utilizing the proposed algorithm, given a PLL dataset (including the employed feature extractor), $\delta_k$, and $\rho_k$ are fixed (or can be estimated on the validation set). We need to choose the appreciate values of $k$ and $\gamma_i$ ($\tau$). Before formally introducing the empirical experiment of evaluating the effects of varying $k$ and $\gamma_i$, let us additionally come up with another conclusion shown below.

**Theorem 2.** *Given the same assumption of the $(k, \delta_k, \rho_k)$-label distinguishability and notations in Theorem 1, when increasing the number of eliminated candidate labels (i.e., $\gamma_i^2 > \gamma_i^1$), the extra pruning error can be bounded by*

$$\mathbb{P}(O_{iy'}^{(\xi_i^2)} < O_{iy}) - \mathbb{P}(O_{iy'}^{(\xi_i^1)} < O_{iy}) \leqslant \sum_{j=1}^{k} \sum_{m=\xi_i^2}^{\xi_i^1 - 1} \binom{|Y_i'|}{m}\eta^m(1-\eta)^{|Y_i'|-m}b_{\delta_k}(k, j), \quad (5)$$

*where $\xi_i^1 = (|Y_i'| - \gamma_i^1 + 1)$, $\xi_i^2 = (|Y_i'| - \gamma_i^2 + 1)$, and other notations are the same as those used in Theorem 1.*

The proof is provided in Appendix C. The Theorem 2 implies that the extra pruning error caused by increasing the number of eliminated candidate labels is also bounded by an upper bound associated with the four key factors. Now, the key challenge is to choose the appreciate values of $k$ and $\gamma_i$ ($\tau$), thereby decreasing the upper bound of the pruning error. To this end, we conduct the numerical simulation experiment to empirically evaluate the effects of varying $k$ and $\gamma_i$ for the upper bound.

**Numerical simulation.** Here, we set various values of $\delta_k$ and $\rho_k$ where a small value of $\delta_k$ ($\rho_k$) implies high-quality representations (candidate label sets of low label ambiguity). First, we aim to explore the effect of varying $k$ on the upper bound. Suppose a PLL instance $x_i$ possesses three candidate labels (i.e., $|Y_i| = 3$) and the number of eliminated candidate label $\gamma_i = 1$, we set various values of $\delta_k = [0.2, 0.4, 0.6, 0.8]$ and $\rho_k = [0.2, 0.8]$. By taking these values into the formulation (4), we can empirically calculate the upper bound as shown in Figure 1. The experiment result shows that under a small $\rho_k$, as $k$ increases, the upper bound generally decreases except for a large $\delta_k$, and under a large $\rho_k$, the upper bound maintain stability only with a small $\delta_k$ when $k$ increases. Moreover, to explore the effect of increasing the number of eliminated candidate labels $\gamma_i$, we assume a PLL instance $x_i$ equipped with a larger size of candidate label set $|Y_i| = 6$, thereby leading to a wider pruning range of $\gamma_i \in [1, 5]$. The experiment result implies that when increasing the number of eliminated candidate labels, the upper bound maintains stability with a small $\delta_k$ and $\rho_k$. Based on these observations, we can draw two empirical conclusions: (1) a small value of $k$ can achieve a relatively low upper bound of the pruning error on imperfect features or high label ambiguity; (2) eliminating more candidate labels is feasible by using high-quality representations under low label ambiguity.

## 4 EXPERIMENTS

In this section, we conduct extensive experiments to verify the effectiveness of the proposed method for *candidate label set pruning* (CLSP). We first introduce the experimental setup and then present comprehensive experimental results. Based on these results, we conduct in-depth empirical analyses.

Table 1: Test accuracy comparison on CIFAR-10 and CIFAR-100 datasets under uniform, label-dependent (LD), and instance-dependent (ID) PLL settings. The row in gray color indicates the PLL method using the *pruned* candidate label set. The better result is highlighted in bold.

| Dataset | $q$ | CC | PRODEN | LWS | CAVL | PiCO | CR | ABLE | IDGP | POP |
|---|---|---|---|---|---|---|---|---|---|---|
| C-10 | 0.4 | 81.67 | 81.11 | 84.85 | 78.14 | 94.20 | 96.99 | 94.53 | 92.34 | 95.19 |
| | | **86.45** | **82.07** | **86.68** | **81.25** | **94.44** | **97.24** | **94.97** | **93.38** | **95.64** |
| | 0.6 | 71.16 | 79.81 | 81.67 | 54.52 | 92.96 | 96.47 | 93.69 | 89.48 | 94.57 |
| | | **84.62** | **82.42** | **85.87** | **80.99** | **94.32** | **97.21** | **94.92** | **92.52** | **95.48** |
| | LD | 89.57 | 81.83 | 86.18 | 80.43 | 94.59 | 97.24 | 94.77 | 92.47 | 95.63 |
| | | **90.81** | **82.49** | **87.54** | **82.17** | **94.49** | **97.58** | **95.19** | **92.82** | **95.87** |
| | ID | 73.92 | 78.03 | 78.70 | 67.21 | 91.08 | 87.89 | 91.17 | 84.45 | 93.63 |
| | | **77.57** | **81.92** | **84.67** | **77.97** | **93.41** | **95.90** | **93.99** | **92.08** | **95.05** |
| C-100 | 0.05 | 64.05 | 48.68 | 51.18 | 41.00 | 72.31 | 83.16 | 74.43 | 68.39 | 76.35 |
| | | **64.36** | **49.72** | **52.69** | **49.62** | **72.66** | **83.53** | **75.08** | **68.86** | **76.85** |
| | 0.1 | 62.31 | 46.26 | 45.57 | 21.34 | 56.80 | 82.51 | 74.80 | 67.62 | 74.38 |
| | | **64.05** | **48.38** | **51.62** | **45.48** | **72.51** | **83.39** | **74.76** | **68.55** | **75.95** |
| | H-0.5 | 63.72 | 29.28 | 51.29 | 48.76 | 72.46 | 82.93 | 74.11 | 68.22 | 74.90 |
| | | **65.56** | **40.92** | **53.40** | **48.88** | **73.10** | **83.38** | **75.59** | **68.53** | **75.32** |
| | ID | 63.06 | 49.83 | 53.18 | 47.35 | 71.04 | 80.76 | 74.04 | 66.71 | 73.36 |
| | | **63.30** | **50.11** | 52.74 | **48.24** | **71.78** | **80.93** | **74.49** | **67.23** | **74.26** |

Table 2: Test and transductive accuracy comparison on VOC dataset. The row in gray color indicates the PLL method using the *pruned* PLL data. The better result is highlighted in bold.

| Dataset | | CC | PRODEN | LWS | CAVL | CRDPLL | SoLar | RECORDS |
|---|---|---|---|---|---|---|---|---|
| VOC | Test | 34.21 | 43.59 | 17.78 | 32.25 | 21.26 | 64.53 | 62.52 |
| | | **50.81** | **47.85** | **28.08** | **49.45** | **38.53** | **65.42** | **65.38** |
| | Trans. | 72.15 | 77.26 | 64.46 | 77.43 | 67.09 | 76.56 | 40.33 |
| | | **77.51** | **79.01** | **71.07** | **86.88** | **75.09** | **82.35** | **68.32** |

## 4.1 EXPERIMENTAL SETUP

**Datasets.** We use three benchmark datasets, i.e., CIFAR-10 (Krizhevsky et al., 2009), CIFAR-100 (Krizhevsky et al., 2009), Tiny-ImageNet (Wu et al., 2017), and a real-world PLL dataset PASCAL VOC (Everingham et al., 2015). Besides, we consider a long-tailed PLL setting (Wang et al., 2022a; Hong et al., 2023) on CIFAR-10 and CIFAR-100 denoted by CIFAR-10-LT and CIFAR-100-LT, respectively. Following the previous work (Hong et al., 2023), we employ the imbalance rate $\phi = [50, 100]$ that denotes the ratio of the example sizes of the most frequent and least frequent classes, i.e., $\phi = N_{(c)}/N_{(1)}$, where $[N_{(c)}, \cdots, N_{(1)}]$ is in the descending order.

**Candidate label generation.** To simulate the real-world generation process of the candidate label set, existing deep PLL research commonly considers various generation models of candidate labels. We consider three general candidate generation models that involve different types of flipping probability: uniform generation, label-dependent (LD) generation, and instance-dependent (ID) generation. Specifically, we consider a uniform probability $q = [0.4, 0.6]$ on CIFAR-10, $q = [0.3, 0.5]$ on CIFAR-10-LT, $q = [0.05, 0.1]$ on CIFAR-100, $q = [0.03, 0.05]$ on CIFAR-100-LT, and $q = [0.01, 0.05]$ on Tiny-ImageNet respectively. For the label-dependent generation, we consider a candidate probability vector $\boldsymbol{q} = [0.5, 0.4, 0.3, 0.2, 0.1]$ for each label on CIFAR-10, and generate hierarchical candidate labels that belong to the same super-class with a probability 0.5 on CIFAR-100. For the instance-dependent generation, following the previous work (Xu et al., 2021), we use the prediction of a neural network trained with original clean labels as the sample- and label-wise flipping probability.

**Evaluation metric.** To evaluate the proposed CLSP method, we can use $\alpha$-error and $\beta$-coverage in Definition 1 as two metrics. The smaller (larger) the value of $\alpha$ ($\beta$), the better the performance. Besides, we also employ the F1 score to evaluate the pruning. Specifically, the precision of pruning: $\text{Precision} = 1 - \alpha$ and the recall of pruning: $\text{Recall} = \beta$. Thus, $F_1 = 2((1 - \alpha)\beta)/(1 - \alpha + \beta)$. The larger the value of the F1 score, the better the performance.

**Feature extractor.** In this paper, we consider various visual feature extractors including visual-only ResNet-18-based models: ResNet-S that performs conventional supervised learning with original clean supervision, ResNet-SSL that leverages self-supervised learning (SimCLR (Chen et al.,

Table 3: Test accuracy comparison on class-imbalanced CIFAR-10 and CIFAR-100 datasets under long-tailed PLL settings. The row in gray color indicates the PLL method using the *pruned* candidate label set. The better result is highlighted in bold.

| Dataset | $q$ | $\phi$ | CC | PRODEN | LWS | CAVL | CR | SoLar | RE |
|---------|-----|--------|-----|--------|-----|------|-----|-------|-----|
| C-10-LT | 0.3 | 50 | 75.31 | 76.73 | 77.28 | 44.18 | 71.53 | 84.48 | 79.95 |
| | | | **77.51** | **78.66** | **78.79** | **44.21** | **78.37** | **84.69** | 79.80 |
| | | 100 | 67.36 | 65.58 | 65.52 | 43.39 | 82.61 | 75.53 | 70.86 |
| | | | **70.48** | **71.23** | **72.49** | **43.41** | **84.35** | **76.82** | **71.43** |
| | 0.5 | 50 | 59.90 | 62.95 | 62.22 | 42.84 | 47.92 | 82.41 | 75.48 |
| | | | **66.85** | **65.30** | **64.13** | **48.36** | **64.94** | **83.57** | **76.48** |
| | | 100 | 55.36 | 55.37 | 57.19 | 44.85 | 60.43 | 71.50 | 65.73 |
| | | | **63.05** | **62.19** | **62.13** | 42.59 | **68.05** | 70.85 | **67.08** |
| C-100-LT | 0.03 | 50 | 43.70 | 43.93 | 43.40 | 32.39 | 45.06 | 48.31 | 39.91 |
| | | | **44.67** | **44.69** | **45.24** | **33.33** | **45.80** | **49.27** | **40.30** |
| | | 100 | 39.32 | 38.71 | 38.07 | 30.55 | 51.24 | 42.76 | 36.42 |
| | | | **40.14** | **39.52** | **39.24** | **31.75** | **52.52** | **43.81** | **37.44** |
| | 0.05 | 50 | 42.37 | 39.53 | 38.89 | 28.43 | 42.92 | 46.39 | 44.82 |
| | | | **42.66** | **41.18** | **40.70** | **29.46** | **44.45** | **47.01** | **46.06** |
| | | 100 | 35.59 | 34.94 | 34.43 | 26.16 | 48.91 | 40.94 | 40.03 |
| | | | **37.52** | **37.16** | **36.31** | **26.76** | **49.75** | **42.33** | **40.44** |

Table 4: The magnitude change of the candidate label set, $\alpha$-error (%), and $\beta$-coverage of the pruning method under various PLL settings. O (vs. P) Avg. C means the average number of examples' candidate labels on the original (vs. pruned) PLL dataset.

| Dataset | CIFAR-10 | | | | CIFAR-100 | | | | Tiny-ImageNet | | | VOC |
|---------|------|------|------|------|------|------|-------|------|------|------|------|------|
| | 0.4 | 0.6 | LD | ID | 0.05 | 0.1 | H-0.5 | ID | 0.01 | 0.05 | ID | × |
| O Avg. C | 4.6 | 6.4 | 2.5 | 3.3 | 6.0 | 10.9 | 3.0 | 4.8 | 3.0 | 11.0 | 8.3 | 2.5 |
| P Avg. C | **2.1** | **2.9** | **1.2** | **2.6** | **2.9** | **5.0** | **1.3** | **4.4** | **2.3** | **7.2** | **7.1** | **1.8** |
| $\alpha$-error (%) | .18 | .16 | .36 | .22 | .43 | .50 | 2.9 | .54 | .46 | .65 | .38 | 5.2 |
| $\beta$-coverage | .69 | .64 | .89 | .32 | .62 | .59 | .82 | .08 | .49 | .37 | .16 | .47 |

2020)) without supervision, ResNet-I that directly uses the pre-trained model on the ImageNet-V1 dataset, and visual encoders in the pre-trained visual-language model: CLIP (Radford et al., 2021), ALBEF (Li et al., 2021), BLIP-2 (Li et al., 2023b). More details are shown in Appendix D.

**Deep PLL methods.** In this paper, we consider eleven state-of-the-art deep PLL methods, including six conventional deep PLL methods, e.g., CC, PRODEN, LWS, CAVL, PiCO, CR, three instance-dependent deep PLL methods ABLE, IDGP, and POP, and two long-tailed deep PLL methods SoLar and RECORDS.

**Implementation.** We employ BLIP-2 as the feature extractor based on the open-source library *LAVIS* (Li et al., 2023a). For $k$-NN searching, we employ *Faiss* (Johnson et al., 2019), a library for efficient similarity search and clustering of dense vectors. The parameters $k$ and $\tau$ used in the proposed algorithm are shown in Table 5. For each PLL method, we keep the same training scheme with both original and pruned candidate label sets. More details are shown in Appendix D.

## 4.2 EXPERIMENTAL RESULTS

**CLSP performance.** As shown in Table 4, we present the average number of candidate labels on the PLL datasets before and after the pruning respectively, as well as the defined $\alpha$-error (%) and $\beta$-coverage of the pruning. As a whole, we can see that the proposed method significantly reduces the size of candidate label sets at relatively low pruning error rates. Specifically, $\alpha$-error (%) of the pruning maintains a relatively low level which is mostly less than 1%, and $\beta$-coverage of the pruning reaches a high level. The largest value of $\alpha$-error is 5.2% on PASCAL VOC. This is because the feature quality of images on the dataset is relatively low due to their complicated visual appearances. The smallest value of $\beta$-coverage is 0.08 on CIFAR-100 (ID). This is because label ambiguity on CIFAR-100 (ID) is knotty (i.e., false candidate labels always co-occur with the true label), which is challenging for the CLSP task. Besides, we also explore the effects of different feature extractors as

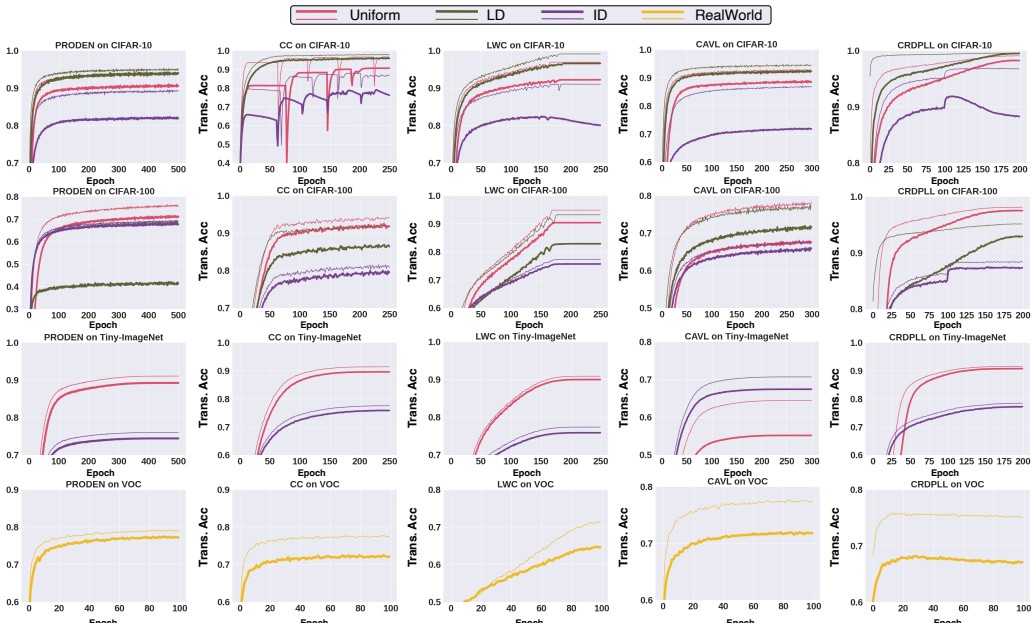

Figure 2: Transductive accuracy of PRODEN, CC, LWC, CAVL, and CRDPLL algorithms on CIFAR-10, CIFAR-100, Tiny-ImageNet, and VOC datasets. *Bold* (vs. *thin*) curves indicate the corresponding algorithm using the *original* (vs. *pruned*) PLL data. Best viewed in color.

shown in Appendix E. On the whole, the performance of pre-trained vision-language models, i.e., CLIP, ALBEF, and BLIP-2, is better than visual-only models ResNet-SSL and ResNet-I.

**PLL performance comparison.** To show the effect of the proposed CLSP method on improving deep PLL methods, we conduct each PLL method on the original and pruned PLL datasets respectively under the same training scheme. From comprehensive experimental results, we can draw the following conclusions:

- *Overall performance improvement.* On the whole, from the results of Tables 1, 3, 2, and 10, we can see that test accuracy of PLL methods under almost all cases (145/149≈97.3%) is significantly improved after training on the pruned PLL dataset. This result directly validates the effectiveness of the proposed CLSP method in improving deep PLL methods. Furthermore, we also present the transductive accuracy of PLL methods in Tables 7, 8, 9 and Figure 2. We can see that the proposed CLSP method also definitely increases the transductive accuracy by a significant margin, which implies that more PLL examples are identified correctly after eliminating their potential false candidate labels. This observation validates our claim that the proposed CLSP method can reduce label ambiguity and boost label disambiguation effectively. Besides, there are a few bad cases (4/149 ≈2.7%) in the experimental results where the performance of certain PLL methods has slightly degraded after training with pruned PLL data. We argue that this is because the involved PLL methods (i.e., ABLE and SoLar) in bad cases have a time-consuming training procedure, i.e., 500 and 1000 epochs respectively, hence they tend to overfit noisy PLL instances eventually, thereby leading to performance degradation.

- *The effect on different candidate generations.* Also, we further analyze the effect of the proposed CLSP method on uniform, LD, and ID generations. Under uniform generation, PLL methods generally have a distinct performance degradation as $q$ increases. For example in Table 1 on CIFAR-10, the average reduced test accuracy of PLL methods from $q = 0.4$ to $q = 0.6$ is about 3.6%, while the average reduced test accuracy of PLL methods using the pruned candidate label set from $q = 0.4$ to $q = 0.6$ is about 0.6%. This validates the effectiveness of the proposed CLSP method to eliminate uniformly generated candidate labels. Particularly, the improvement is more significant under LD and ID generations. This implies that the proposed CLSP method is superior against knotty candidate labels in LD and ID cases.

- *The effect on LT data distribution.* As shown in Table 3, the performance of PLL methods has a serious performance degradation. This is because the class-imbalanced data affects label disam-

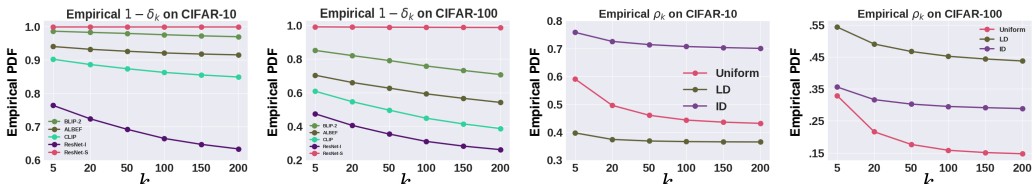

Figure 3: Empirically calculated values of $\delta_k$ and $\rho_k$ under various settings and feature extractors.

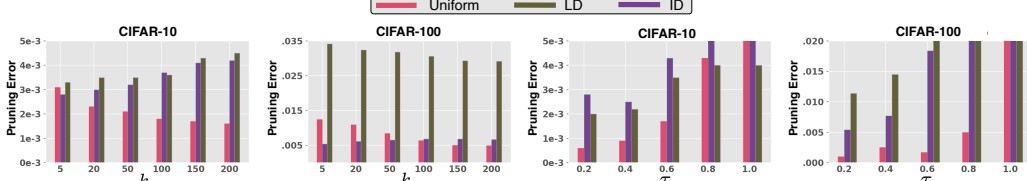

Figure 4: Various values of $k$ and $\tau$ on CIFAR-10 and CIFAR-100 under different PLL settings.

biguation greatly. We can see that PLL methods have a significant performance improvement by training with the pruned PLL data.

- *The significant improvement on the real-world dataset.* Note that the PASCAL VOC dataset is a real-world class-imbalanced PLL dataset (Hong et al., 2023). As shown in Table 2, the proposed CLSP method significantly improves the transductive and test accuracy of PLL methods. This shows the practical value of the proposed CLSP method in real-world tasks.

**Parameter analysis.** Since the values of $\delta_k$ and $\rho_k$ can be empirically calculated by $\delta_k = 1 - (\sum_{i=1}^{n} \sum_k \mathbb{I}(y_i = y_i^{(k)}))/kn$ and $\rho_k = (\sum_{i=1}^{n} \max_{y_i' \in Y_i'} \sum_k \mathbb{I}(y_i' = y_i'^{(k)}))/kn$, where $Y_i' = Y_i \backslash \{y_i\}$ is the set of false candidate labels. Here, we show the empirical values of $\delta_k$ and $\rho_k$ on CIFAR-10 and CIFAR-100. As shown in Figure 3, the overall trend of $1 - \delta_k$ and $\rho_k$ is decreasing as $k$ increases. The better the feature extractor, the larger the value of $\delta_k$, indicating high-quality representations. Figure 4 shows the results of various values of $k$ and $\tau$ on CIFAR-10 and CIFAR-100 under different PLL settings. From Figure 4, we can find that both the ID and LD candidate label generation processes lead to a larger value of $\rho_k$ than the uniform case, which implies a higher label ambiguity. We employ BLIP-2 as the feature extractor with a large $\delta_k$ and thus are able to combat the ID and LD cases. Based on the empirical observation, we further evaluate various values of $k$ and $\tau$. In particular, we evaluate one parameter with another fixed. As shown in Figure 4, when the number of $k$-NN increases, the pruning error on uniform generation drops significantly at first and maintains stability under a large $k$. This phenomenon accords with the theoretical analysis that a small $\delta_k$ favors a large $k$. On the contrary, the pruning error under LD and ID candidate generations increases obviously as $k$ increases. This means a small $k$ is enough under a large $\rho_k$. As for the parameter $\tau$, we can see that increasing the value of $\tau$ consistently increases the pruning error under all cases. Fortunately, the pruning error growth under uniform candidate generation is relatively slow, and thus it is favorable to select a relatively large $\tau$.

## 5 CONCLUSION

In this paper, we pioneer a new PLL-related task called candidate label set pruning (CLSP) that aims to reduce the size of candidate label sets of PLL instances. To this end, we propose the first CLSP method that eliminates certain potential false candidate labels of PLL instances based on a "down-vote" statistic from their $k$-NN instances in the representation space. Theoretically, we analyze the effects of the representation quality and label ambiguity against the upper bound of the pruning error. Empirically, extensive experiments on both benchmark-simulated and real-world PLL datasets validate the superiority of the proposed CLSP method to significantly improve state-of-the-art deep PLL methods. In the future, it is also interesting to develop more effective methods for CLSP. We hope our work will draw more attention of the PLL community from the learning-centric perspective toward the data-centric perspective.

ACKNOWLEDGMENTS

Lei Feng is supported by the National Natural Science Foundation of China (Grant No. 62106028) and the Chongqing Overseas Chinese Entrepreneurship and Innovation Support Program. Guowu Yang is supported by the National Natural Science Foundation of China (Grant No. 62172075). This paper is partially supported by the Sichuan Science and Technology Program (Grant No. 2022YFG0189).

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

## A  THE PSEUDO-CODE OF THE PROPOSED ALGORITHM

---

**Algorithm 1:** The proposed CLSP method

---

**Input:** A PLL dataset $\mathcal{D} = \{(\boldsymbol{x}_i, Y_i)\}_{i=1}^n$, a feature extractor $\Phi(\cdot)$, parameters $k, \tau$.
**Output:** The pruned PLL dataset $\bar{\mathcal{D}}$
  // Feature Extract
1 Obtain feature representations of all instances the feature extractor by $\Phi(\cdot)$;
  // Pruning towards each instance
2 **for** $i \leqslant n$ **do**
     // $k$-NN searching using *faiss*
3      Search $k$-NN instances $[(\boldsymbol{x}_i^{(1)}, Y_i^{(1)}), \cdots, (\boldsymbol{x}_i^{(k)}, Y_i^{(k)})]$ on $\mathcal{D}$ for each instance $(\boldsymbol{x}_i, Y_i)$;
     // Down-voting from $k$-NN instances
4      Calculate a down-vote statistic $O_i$ towards each instance $\boldsymbol{x}_i$ by Eq. (1);
     // Counting
5      Count the number of eliminated candidate labels $\gamma_i$ towards each instance$\boldsymbol{x}_i$ by Eq. (2);
     // Pruning each candidate label set
6      Select the eliminated candidate labels $\widetilde{Y}_i$ based on the highest down-voting by Eq. (3);
     // Obtain the pruned candidate label set
7      Obtain the pruned candidate label set towards the instance $\boldsymbol{x}_i$: $\bar{Y}_i = Y_i \backslash \widetilde{Y}_i$.
8 **end**
9 Return the pruned PLL dataset $\bar{\mathcal{D}} = \{(\boldsymbol{x}_i, \bar{Y}_i)\}_{i=1}^n$;

---

As shown in Algorithm 1, the proposed method does not involve the complex and time-consuming training process. The most time-consuming step is $k$-NN searching where we utilize the open-source library *Faiss* to accelerate the process. The pruned PLL dataset can be directly leveraged in existing deep PLL methods.

## B  THE PROOF OF THEOREM 1

Recall that in Section 3.3, there are one random variable $O_{iy} \sim B(k, 1-t)$ and $(|Y_i'|)$ *i.i.d.* random variables $O_{iy'} \sim B(k, 1-q)$. The number of eliminated candidate labels $\gamma_i \in [1, |Y_i'|]$. The event of incorrect pruning for the instance $\boldsymbol{x}_i$ is $[O_{iy'}^{(\xi_i)} < O_{iy}]$ where $\xi_i = |Y'| - \gamma_i + 1$ and $O_{iy'}^{(k)}$ is the $k$-th *order statistic* $[O_{iy'}^{(1)}, \cdots, O_{iy'}^{(k)}, \cdots, O_{iy'}^{(|Y_i'|)}]$. Now, we derive an upper bound for the probability of getting incorrect pruning for the instance $\boldsymbol{x}$:

$$\mathbb{P}(O_{iy'}^{(\xi_i)} < O_{iy}) = \sum_{j=0}^k P(O_{iy'}^{(\xi_i)} < O_{iy}|O_{iy} = j)P(O_{iy} = j)$$

$$= \sum_{j=1}^k P(O_{iy'}^{(\xi_i)} \leqslant j - 1)P(O_{iy} = j)$$

$$\leqslant \sum_{j=1}^k \sum_{m=\xi_i}^{|Y_i'|} \binom{|Y_i'|}{m} \eta^m (1-\eta)^{(|Y_i'|-m)} b_{\delta_k}(k, j),$$

where $\xi_i = (|Y_i'| - \gamma_i + 1)$, $\binom{n}{r} = \frac{n!}{r!(n-r)!}$ is the combination formula, $b_{\delta_k}(k, j) = \binom{k}{j} \delta_k^j (1-\delta_k)^{k-j}$ denotes the probability mass function of a binomial distribution $B(k, \delta_k)$, and $\eta = I_{\rho_k}(k-j+1, j)$ where $I_{\rho_k}(k, j) = \int_0^{\rho_k} t^{k-1}(1-t)^{j-1}dt$ is the regularized incomplete beta function.

## C  THE PROOF OF THEOREM 2

Based on the notations in the proof of Theorem 1, we further denote $\gamma_i^1$ and $\gamma_i^2$ by two numbers of eliminated candidate labels respectively satisfying $\gamma_i^1 < \gamma_i^2$. Then we can derive an upper bound for

Table 5: The values of parameters $k$ and $\tau$ used in the proposed method under various PLL settings.

| Setup | Uniform | | | LD | | ID | | | LT | | $\times$ |
|---|---|---|---|---|---|---|---|---|---|---|---|
| | C-10 | C-100 | T-I | C-10 | C-100 | C-10 | C-100 | T-I | C-10 | C-100 | VOC |
| $\tau$ | 0.6 | 0.6 | 0.4 | 0.6 | 0.6 | 0.2 | 0.2 | 0.2 | 0.2 | 0.2 | 0.1 |
| $k$ | 150 | 150 | 150 | 50 | 150 | 5 | 5 | 50 | 50 | 50 | 5 |

the extra pruning error

$$
\mathbb{P}(O_{iy'}^{(\xi_i^2)} < O_{iy}) - \mathbb{P}(O_{iy'}^{(\xi_i^1)} < O_{iy}) = \sum_{j=0}^{k} P(O_{iy'}^{(\xi_i^2)} < O_{iy}|O_{iy} = j)P(O_{iy} = j)
$$

$$
- \sum_{j=0}^{k} P(O_{iy'}^{(\xi_i^1)} < O_{iy}|O_{iy} = j)P(O_{iy} = j)
$$

$$
\leqslant \sum_{j=1}^{k} \sum_{m=\xi_i^2}^{|Y_i'|} \binom{|Y_i'|}{m} \eta^m (1-\eta)^{(|Y_i'|-m)} b_{\delta_k}(k, j)
$$

$$
- \sum_{j=1}^{k} \sum_{m=\xi_i^1}^{|Y_i'|} \binom{|Y_i'|}{m} \eta^m (1-\eta)^{(|Y_i'|-m)} b_{\delta_k}(k, j)
$$

$$
\leqslant \sum_{j=1}^{k} \sum_{m=\xi_i^2}^{\xi_i^1-1} \binom{|Y_i'|}{m} \eta^m (1-\eta)^{|Y_i'|-m} b_{\delta_k}(k, j),
$$

where $\xi_i^1 = (|Y_i'| - \gamma_i + 1)$, $\xi_i^2 = (|Y_i'| - \gamma_i + 1)$, and other notations' definitions are the same as that in Theorem 1.

## D  EXPERIMENTAL CONFIGURATION

In this section, we further show the detailed experimental configuration of the proposed method and partial-label learning (PLL) methods. First, we present the values of parameters $k$ and $\tau$ used in the proposed method on each dataset under different partial-label learning (PLL) settings.

**Candidate label generation.**    Formally, given an example $(\boldsymbol{x}_i, y_i) \sim p(\boldsymbol{x}, y)$, candidate generation models intrinsically consider the flipping probability of a false label $y_j' \neq y_i \cap y_j' \in \mathcal{Y}$ being a candidate one. (1) *Uniform Generation* (Lv et al., 2020): this case is a naive simple generation model that assumes each domain label has the same flipping probability of being a candidate. The uniform real-value flipping probability is $P(\widetilde{y} \in Y|y) = q$; (2) *Label-dependent Generation* (Wang et al., 2022b): instead of treating all domain classes equally, semantic-similar domain labels against the true label are more likely to be inside a candidate label set. The label-wise flipping probability is $P(\widetilde{y}_j \in Y|y) = q_j$; (3) *Instance-dependent Generation* (Xu et al., 2021): the above two cases do not consider the instance itself. In a more realistic scenario, candidate labels are usually generated according to the specific features of the instance. The sample- and label-wise flipping probability is $P(\widetilde{y}_{ij} \in Y_i|\boldsymbol{x}_i, y_i) = q_{ij}$. Specifically, we consider a uniform probability $q = [0.4, 0.6]$ on CIFAR-10, $q = [0.3, 0.5]$ on CIFAR-10-LT, $q = [0.05, 0.1]$ on CIFAR-100, $q = [0.03, 0.05]$ on CIFAR-100-LT, and $q = [0.01, 0.05]$ on Tiny-ImageNet respectively. For the label-dependent generation, we consider a candidate probability vector $\boldsymbol{q} = [0.5, 0.4, 0.3, 0.2, 0.1]$ for each label on CIFAR-10, and generate hierarchical candidate labels that belong to the same super-class with a probability 0.5 on CIFAR-100. For the instance-dependent generation, following the previous work (Xu et al., 2021), we use the prediction of a neural network trained with original clean labels as the sample- and label-wise flipping probability $q_{ij}$.

**Deep PLL methods.**    CC (Feng et al., 2020): a classifier-consistent method that assumes a set-level uniform data generation process; PRODEN (Lv et al., 2020): a self-training-based method that progressively identifies the true labels using the output of the classifier itself; LWS (Wen et al., 2021): a set of loss functions that weights the risk function by means of a trade-off between losses on candidates and non-candidates; CAVL (Zhang et al., 2022): a method motivated by Class Activation Map (CAM) and using the corresponding class activation value for label disambiguation; PiCO (Wang et al., 2022b): a contrastive learning-based method that establishes class prototypes for label

disambiguation; CRPLL (Wu et al., 2022): a regularization based method that achieves state-of-the-art performance in instance-independent PLL; ABLE (Xia et al., 2022): an ambiguity-induced contrastive learning method that leverages label information into contrastive learning; IDGP (Qiao et al., 2023): a maximum a posterior (MAP) based probabilistic method that considers a compositional generation process of candidates; SoLar (Wang et al., 2022a): an optimal transport-based framework that refines the disambiguated labels towards matching the marginal class prior distribution; RECORDS (Hong et al., 2023): a dynamic rebalancing method that is benign to the label disambiguation process and theoretically converges to the oracle class prior; POP (Xu et al., 2023b): a theoretically grounded algorithm to filter out false candidate labels progressively in every epoch.

**More details about the $k$-NN algorithm and feature extractors.** We employ the visual encoder of BLIP-2 to extract 768-dimensional high-quality representations for all training PLL instances. Then, we leverage *faiss* to conduct the fast $k$-NN searching for each instance based on the squared Euclidean Distance. Besides, we additionally train two visual-only ResNet-18-based models (i.e., ResNet-SSL, ResNet-S) on each PLL dataset. Note that we did not use them in the proposed method and just used them for comparing the effects of different types of feature extractors. Specifically, ResNet-S is trained with the original clean supervision using the cross-entropy loss (note that we do not employ data augmentation techniques), while ResNet-SSL is trained by the self-supervised learning method SimCLR without any supervision. The weak and strong data augmentations used in SimCLR follow the original configurations on the corresponding PLL dataset. In addition, ResNet-I is directly employed by loading the checkpoint "IMAGENET1K_V1". The dimension of these three models' output representations is 512.

**Parameters $k$ and $\tau$ in the proposed method.** As shown in Table 5, based on our theoretic analysis for the effect of $k$ and $\gamma$ ($\tau$) against the upper bound Eq. (4) in Theorem 1, we commonly select a large $k$ and $\tau$ on CIFAR-10 and CIFAR-100 datasets under uniform generation due to the high-quality extracted features on CIFAR-based datasets and the low label ambiguity under uniform generation, while we choose a relatively low $k$ and $\tau$ under ID generation due to the knotty label ambiguity, or on VOC due to the relatively low-quality extracted features of examples in the dataset. The effectiveness of the theoretic-inspired guidance is validated in the parameter analysis shown in Figure 4 where the appropriate values of parameters $k$ and $\tau$ effectively reduce the pruning error.

**Training scheme of PLL methods.** Besides, we further present the detailed training scheme of PLL methods including model architecture, learning rate, learning rate scheduler, weight decay, batch size, and data augmentation. On the whole, we employ a base training scheme: a ResNet-18 model, learning rate is 1e-2, and weight decay is 1e-3. On CIFAR-10 and CIFAR-100, CC, PRO-DEN, LWS, and CAVL do not employ a learning rate scheduler and the data augmentation technique which is the same as the original implementation. But, on more difficult datasets CIFAR-10-LT, CIFAR-100-LT, Tiny-ImageNet, and VOC, they are equipped with a consistency regularization with augmented examples and a "CosineAnnealingLearningRate" scheduler which is a scheduling technique that starts with a very large learning rate and then aggressively decreases it to a value near 0 before increasing the learning rate again. Especially, on VOC, the epoch is set to 100 for all PLL methods to avoid overfitting. Although there are different configurations for PLL methods, they both employ the same training scheme for the original and pruned candidate label sets, which ensures the performance improvement comes from only the effect of pruning.

**PASCAL VOC.** Following the previous work (Hong et al., 2023), we construct the dataset where objects in images are cropped as instances and all objects appearing in the same original image are regarded as the labels of a candidate set. As for the characteristics of PASCAL VOC, the number of classes is 20, the number of training (test) instances is 11706 (4000), and the average number of candidate labels is 2.46. Particularly, the imbalance ratio is 118.8.

# E    MORE EXPERIMENTAL RESULTS

In this section, we present additional experiment results including the F1 score of using different feature extractors, transductive accuracy comparison, and training loss curves of PLL methods.

---

https://github.com/salesforce/LAVIS
https://github.com/facebookresearch/faiss
https://github.com/google-research/simclr

Table 6: F1 score (%) of different feature extractors adopted in the proposed algorithm.

| Dataset | $q$ | ResNet-S | ResNet-SSL | ResNet-I | CLIP | ALBEF | BLIP-2 |
|---|---|---|---|---|---|---|---|
| CIFAR-10 | 0.4 | **91.71** | 79.47 | 89.55 | 90.82 | 91.14 | 91.61 |
| | 0.6 | **90.05** | 76.98 | 87.87 | 89.03 | 89.42 | 89.97 |
| | LD | **97.74** | 78.91 | 91.94 | 95.63 | 96.33 | 97.48 |
| | ID | 70.41 | 63.73 | 68.03 | 69.62 | 69.85 | **70.50** |
| CIFAR-100 | 0.05 | **89.16** | 71.28 | 87.42 | 88.07 | 88.39 | 89.08 |
| | 0.1 | **87.87** | 67.11 | 85.55 | 86.35 | 86.86 | 87.69 |
| | H-0.5 | **96.00** | 69.93 | 86.43 | 89.07 | 90.70 | 93.89 |
| | ID | **34.35** | 16.85 | 30.39 | 31.31 | 31.64 | 32.53 |
| Tiny-ImageNet | 0.01 | 79.87 | 70.53 | 82.40 | 82.71 | 82.66 | **82.97** |
| | 0.05 | 71.25 | 62.04 | 73.58 | 74.05 | 74.00 | **74.57** |
| | ID | 48.22 | 45.72 | 49.20 | 49.01 | **49.07** | 49.05 |
| VOC | $\times$ | **86.33** | 73.48 | 78.10 | 79.93 | 80.05 | 78.99 |

Table 7: Transductive accuracy comparison on CIFAR-10 and CIFAR-100 datasets under uniform, label-dependent (LD), and instance-dependent (ID) PLL settings. The row in gray color indicates the PLL method using the *pruned* candidate label set. The better result is highlighted in bold.

| Dataset | $q$ | CC | PRODEN | LWS | CAVL | PiCO | CR | ABLE | IDGP |
|---|---|---|---|---|---|---|---|---|---|
| C-10 | 0.4 | 90.56 | 90.54 | 92.19 | 88.52 | 96.10 | 97.02 | 96.10 | 92.86 |
| | | **96.56** | **93.77** | **96.90** | **92.70** | **97.23** | **99.06** | **98.15** | **96.90** |
| | 0.6 | 80.02 | 86.98 | 86.94 | 61.33 | 94.51 | 96.52 | 93.72 | 82.12 |
| | | **93.96** | **92.66** | **94.99** | **91.04** | **96.56** | **98.53** | **97.16** | **95.57** |
| | LD | 95.87 | 93.87 | 96.55 | 92.23 | 97.42 | 99.49 | 98.41 | 96.43 |
| | | **97.83** | **94.88** | **99.09** | **94.42** | **97.84** | **99.54** | **99.35** | **97.38** |
| | ID | 76.66 | 81.98 | 80.13 | 71.78 | 91.56 | 88.26 | 90.52 | 85.64 |
| | | **86.57** | **89.20** | **91.02** | **86.75** | **94.95** | **96.26** | **94.81** | **92.81** |
| C-100 | 0.05 | 91.81 | 76.22 | 90.34 | 67.54 | 93.23 | 97.49 | 95.57 | 86.12 |
| | | **93.97** | **78.19** | **94.80** | **77.79** | **94.52** | **98.09** | **97.10** | **89.53** |
| | 0.1 | 87.62 | 71.03 | 79.96 | 38.14 | 72.81 | 95.38 | 92.30 | 84.32 |
| | | **91.00** | **75.94** | **89.86** | **72.67** | **92.25** | **96.39** | **94.76** | **85.88** |
| | H-0.5 | 86.50 | 41.31 | 82.83 | 71.39 | 86.39 | 92.90 | 89.17 | 82.73 |
| | | **91.89** | **84.74** | **93.15** | **76.82** | **90.57** | **95.11** | **94.51** | **84.52** |
| | ID | 79.42 | 67.76 | 75.64 | 65.67 | 82.26 | 81.18 | 84.01 | 76.34 |
| | | **81.09** | **69.17** | **77.30** | **67.48** | **83.61** | **81.21** | **85.33** | **77.21** |

**Different feature extractors.** As shown in Table 6, we present the F1 score using different feature extractors including visual-only models ResNet-S, ResNet-SSL, and ResNet-I and vision-language pre-trained models CLIP, ALBEF, and BLIP-2. We can see that ResNet-S achieves the best performance. This is reasonable but impractical since it uses clean supervision. On the whole, feature extractors of CLIP, ALBEF, and BLIP-2, achieve better performance than visual-only ResNet-SSL and ResNet-I models. This shows the powerful visual representation ability of multi-modal models.

**Transductive accuracy comparison.** As shown in Table 7 on CIFAR-10 and CIFAR-100 datasets and Table 8 on CIFAR-10-LT and CIFAR-100-LT datasets and Table 9 on Tiny-ImageNet, we can see that the performance is improved in almost all cases (147/149≈98.7%). Moreover, the improvement in transductive accuracy is more significant than that in test accuracy. These observations definitely validate that the proposed CLSP method greatly boosts label disambiguation of PLL methods. Besides, we further discover that naive PLL methods (e.g., CC, PRODEN, LWS, and CAVL) have more significant performance improvements than advanced PLL methods (e.g., PiCO, CRDPLL, ABLE, and IDGP). We think that this is because naive PLL methods have a limited capability in label disambiguation compared with advanced PLL methods, hence the proposed CLSP method has a bigger effect on promoting label disambiguation in naive PLL methods.

**Training loss curves.** As shown in Figure 5, we present training loss curves of PLL methods on CIFAR-10, CIFAR-100, Tiny-ImageNet, and VOC datasets. Note that *Bold* (*thin*) curves indicate

Table 8: Transductive accuracy comparison on class-imbalanced CIFAR-10 and CIFAR-100 datasets under long-tailed PLL settings. The row in gray color indicates the PLL method using the *pruned* candidate label set. The better result is highlighted in bold.

| Dataset | $q$ | $\phi$ | CC | PRODEN | LWS | CAVL | CR | SoLar | RE |
|---------|-----|--------|-------|--------|-------|-------|-------|-------|-------|
| C-10 | 0.3 | 50 | 94.57 | 95.16 | 95.10 | 86.37 | 96.82 | 98.65 | 85.18 |
| | | | **95.57** | **96.13** | **96.02** | **86.88** | **98.26** | **98.87** | **87.89** |
| | | 100 | 94.86 | 95.05 | 94.95 | 89.47 | 97.09 | 97.63 | 78.26 |
| | | | **96.36** | **96.65** | **96.58** | **90.20** | **98.04** | **98.12** | **82.23** |
| | 0.5 | 50 | 89.33 | 90.85 | 90.57 | 83.74 | 91.47 | 96.22 | 79.51 |
| | | | **91.53** | **92.30** | **92.18** | **86.46** | **95.35** | **97.45** | 78.33 |
| | | 100 | 90.68 | 91.68 | 91.72 | 87.03 | 91.62 | 95.43 | 73.05 |
| | | | **93.22** | **93.95** | **93.58** | **88.15** | **94.23** | **96.65** | 71.12 |
| C-100 | 0.03 | 50 | 92.62 | 91.85 | 91.84 | 83.20 | 95.45 | 94.72 | 87.42 |
| | | | **93.81** | **93.58** | **93.43** | **85.12** | **96.41** | **95.90** | **88.48** |
| | | 100 | 92.54 | 91.76 | 91.46 | 86.17 | 95.61 | 93.56 | 87.97 |
| | | | **93.97** | **93.41** | **93.20** | **87.04** | **96.38** | **94.82** | **88.57** |
| | 0.05 | 50 | 89.10 | 86.63 | 86.57 | 77.38 | 92.50 | 91.23 | 90.30 |
| | | | **90.29** | **88.63** | **88.54** | **79.61** | **93.72** | **91.86** | **91.33** |
| | | 100 | 88.63 | 86.98 | 87.28 | 79.15 | 92.28 | 90.62 | 90.18 |
| | | | **90.29** | **89.19** | **89.14** | **79.61** | **93.72** | **92.21** | **91.51** |

Table 9: Transductive accuracy comparison on Tiny-ImageNet datasets under uniform and instance-dependent PLL settings. The row in gray color indicates the PLL method using the *pruned* candidate label set. The better result is highlighted in bold.

| Dataset | $q$ | CC | PRODEN | LWS | CAVL | CRDPLL |
|---------|-----|-------|--------|-------|-------|--------|
| Tiny-ImageNet | 0.01 | 96.93 | 96.83 | 97.08 | 96.57 | 97.49 |
| | | **97.66** | **97.62** | **97.75** | **97.45** | **98.03** |
| | 0.05 | 89.56 | 89.21 | 89.96 | 55.14 | 90.70 |
| | | **91.36** | **91.04** | **90.84** | **77.09** | **92.31** |
| | ID | 75.81 | 74.41 | 75.86 | 67.39 | 77.19 |
| | | **77.50** | **75.93** | **77.36** | **70.65** | **78.43** |

the corresponding PLL method with the *original* (*pruned*) candidate label set. We can see that PLL methods generally have a faster convergence by using the pruned candidate label set. This phenomenon also discloses another characteristic of the pruning that accelerates the convergence of PLL methods. Generally, the training loss values of PLL methods tend to be smaller after training with pruned data. However, we found an unusual phenomenon where the training loss values of many PLL methods become larger on PASCAL VOC after training with pruned PLL data. We reckon that there are two reasons. Firstly, there exists a higher proportion of noisy PLL instances in PASCAL VOC (where the pruning error is about 5.2%) than in CIFAR and Tiny-ImageNet (where the pruning error is less than 1% in most cases). Secondly, label disambiguation of training PLL instances in PASCAL VOC is more challenging due to the complicated visual objects in PASCAL VOC. Hence, fitting noisy PLL instances in PASCAL VOC is more difficult, thereby leading to a larger training loss value. Notably, the performance of the involved PLL methods on PASCAL VOC still has significant improvements, which implies this phenomenon may be *benign* for PLL methods.

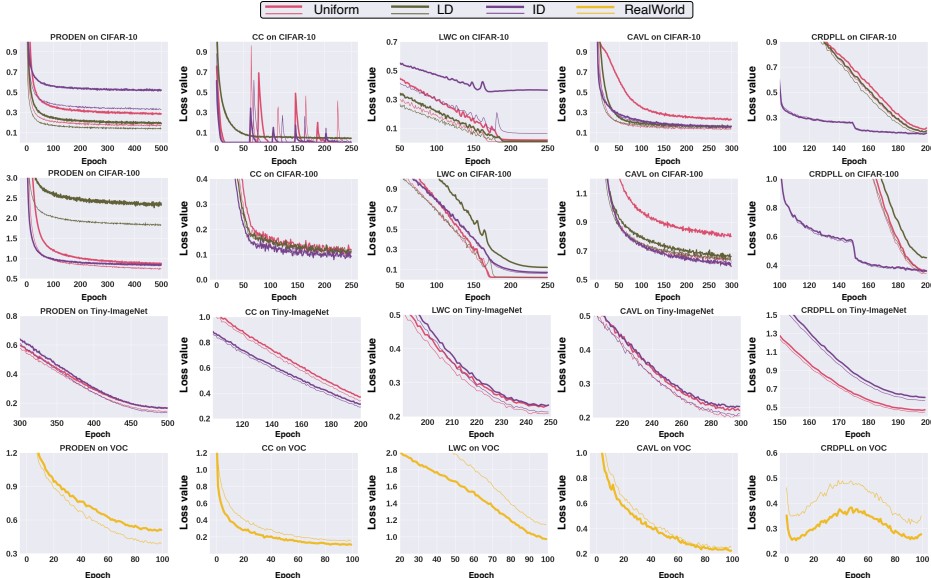

Figure 5: Training loss curves of PRODEN, CC, LWC, CAVL, and CRDPLL methods on CIFAR-10, CIFAR-100, Tiny-ImageNet, and VOC datasets. *Bold* (*thin*) curves indicate the corresponding method with the *original* (*pruned*) candidate label set. Best viewed in color.

Table 10: Test accuracy comparison on Tiny-ImageNet dataset under uniform and instance-dependent PLL settings. The row in gray color indicates the PLL method using the *pruned* candidate label set. The better result is highlighted in bold.

| Dataset | $q$ | CC | PRODEN | LWS | CAVL | CRDPLL |
|---|---|---|---|---|---|---|
| | 0.01 | 65.04 | 65.21 | 66.92 | 64.97 | 67.48 |
| | | **65.35** | **65.28** | **66.98** | **65.43** | **67.56** |
| Tiny-ImageNet | 0.05 | 63.06 | 63.02 | 64.34 | 35.53 | 65.99 |
| | | **63.42** | **63.55** | **65.61** | **52.29** | **66.21** |
| | ID | 61.06 | 59.12 | 61.56 | 53.52 | 63.70 |
| | | **62.11** | **60.15** | **62.30** | **55.95** | **64.27** |