# OpenReview forum: "Candidate Label Set Pruning: A Data-centric Perspective for Deep Partial-label Learning"
_ICLR.cc/2024/Conference — ICLR 2024 oral_

### Official Review · Reviewer_i6zq · 2023-10-18

**Soundness:** 4 excellent
**Presentation:** 4 excellent
**Contribution:** 4 excellent
**Rating:** 8
**Confidence:** 4

**Summary:**

This paper present a novel insight into PLL from the perspective of candidate label set pruning (CLSP), and propose the first CLSP method based on a "down-vote" kNN. The authors also theoretically analyze the effects of the feature quality and label ambiguity against the pruning error. Extensive experiments are conducted on various datasets to validate the superiority of CLSP.

**Strengths:**

This paper is fresh to PLL community from a data prunning perspective, and propose the first candidate label set pruning (CLSP) method based on kNN. The theorectical analysis of the prunning error is reasonable and the comprehensive experimental results validate the effectiveness of the proposed method. The paper is well organized and the expressions are clear. This work is excellent, and will inspire many researchers in PLL community.

**Weaknesses:**

No big problem, only some minor issues, such as, typo, vague expression, and missing reference.

**Questions:**

1. A small typo? I guess the author may forget to divide n in the formulation of \beta in Definition 1, as \beta does not equal to 1 but n in the optimal pruning case. Can author clarify that?
2. It is vague to describe PLL in the abstract that "Partial-label learning (PLL) allows each training example to be equipped with a
set of candidate labels." Does the author intentionally ignore the assumption that "only one is the ground-truth label"? It is better to clarify as some PLL research wave the limitation to investigate a new PLL task (called Unreliable or Noisy PLL).
3. Some state-of-art PLL methods are missing in the reference, such as, A Unifying Probabilistic Framework
for Partially Labeled Data Learning; Mutual Partial Label Learning with Competitive Label Noise.

---

> ### Author Response · Authors · 2023-11-18
> **Response to Reviewer i6zq**
>
> Thank you so much for your meticulous review!
>
> **Q1: A small typo? I guess the author may have forgotten to divide n in the formulation of $\beta$ in Definition 1, as $\beta$ does not equal 1 but n in the optimal pruning case. Can the author clarify that?**
>
> **A:** Thanks for pointing out this issue. We have corrected the issue in the formulation of $\beta$ to improve the clarity: $\beta=\frac{\sum _{i=1} ^{n}|\widetilde{Y} _{i}|}{\sum _{i=1} ^{n}(|Y _{i}|-1)}$. In the optimal pruning case, $\beta=1$ satisfies our consideration.
>
> **Q2: It is vague to describe PLL in the abstract that "Partial-label learning (PLL) allows each training example to be equipped with a set of candidate labels." Does the author intentionally ignore the assumption that "only one is the ground-truth label"? It is better to clarify as some PLL research wave the limitation to investigate a new PLL task (called Unreliable or Noisy PLL).**
>
> **A:** Thanks for pointing out this issue. We would like to explain that this assumption was formally introduced in the definition of PLL in Section 3.1 where each candidate label set $Y_{i}$ consists of a true label $y_{i}$ and a false candidate label set $Y_{i}^{\prime}=Y_{i}\backslash \{y_{i}\}$. To further improve the clarity of our paper, we have highlighted this assumption both in the abstract and problem setup in our paper.
>
> **Q3: Some state-of-art PLL methods are missing in the reference, such as A Unifying Probabilistic Framework for Partially Labeled Data Learning; Mutual Partial Label Learning with Competitive Label Noise.**
>
> **A:** Thank you for letting us know about the related work! We have introduced them [1, 2] in the related work section of our paper.
>
> **References:**
>
> [1] A Unifying Probabilistic Framework for Partially Labeled Data Learning. TPAMI. 2022.
>
> [2] Mutual Partial Label Learning with Competitive Label Noise. ICLR. 2023.

---

> > ### Comment · Reviewer_i6zq · 2023-11-20
> >
> > Thanks for response. I still have one doubt: This work assumes that "the size of the candidate label set are excessively large",  how did the author deal with some instances where the candidate labels are not large at all, say, only 1 candidate label, how to do prunning?

---

> ### Author Response · Authors · 2023-11-20
>
> Thanks for raising this concern. We have already considered this issue in our proposed method. When a PLL instance only has one candidate label, i.e., $|Y_{i}|=1$, the number of eliminated candidate labels $\gamma_{i}=\lceil \tau(|Y_{i}|-1) \rceil$ in Eq. (2) would be always equal to zero, implying that the proposed method would not prune any candidate label of the instance, which is reasonable because the only candidate label is exactly the true label.

---

> > ### Comment · Reviewer_i6zq · 2023-11-22
> >
> > The authors addressed all my concerns, I'll keep score unchanged. Thanks.

---

### Official Review · Reviewer_289c · 2023-10-24

**Soundness:** 3 good
**Presentation:** 3 good
**Contribution:** 3 good
**Rating:** 8
**Confidence:** 4

**Summary:**

This paper focuses on partial label learning, a paradigm of weakly supervised learning, and proposes a training-free method that prunes candidate label sets based on the inconsistency between the representation space and the candidate label space. In particular, when examining each potential label associated with a training instance, if it is not among the candidate labels of the instance's closest neighbors in the feature space, there is a notable likelihood of it being an erroneous label. Theoretically, it provides an upper bound of the per-example pruning error rate and analyzes how the representation quality affects the proposed algorithm.

**Strengths:**

Strengths:
1. Different from the previous learning-centric PLL methods, the method of this paper is proposed from a data-centric perspective, which is novel.
2. The paper theoretically provides an upper bound of the per-example pruning error rate and analyzes how the representation quality affects the proposed algorithm, which is solid.
3. The proposed method is easy to understand and implement.
4. The paper conducts extensive experiments on various settings of PLL.

**Weaknesses:**

Weaknesses:
1. My major concern is that the proposed method will transform a PLL problem into an UPLL problem, which is more challenging due to the existence of the correct label may not be guaranteed in the candidate label set. Although it provides an upper bound of the per-example pruning error rate, the negative impact of eliminating the correct label from the candidate label set is still unknown.
2. The proposed method is dependent on the KNN algorithm, which should be given more details in the main body of the paper. For example, it could be found in the appendix that the KNN algorithm are implemented on the output of a feature extractor. However, what the feature extractor comes from is unknown.
3. [1] also attempts to filter out the incorrect candidate labels, which is suggested to be considered in related works, and even experiments.


[1] Xu, Ning, et al. "Progressive purification for instance-dependent partial label learning." International Conference on Machine Learning. PMLR, 2023.

**Questions:**

1.What is the main difference between a data-centric method and a pre-processing method?

2.Does the feature extractor in the KNN algorithm come from the classifier during the training process? If yes, it seems unreasonable to say that the method is training-free. If not, a pre-trained model should be introduced.

---

> ### Author Response · Authors · 2023-11-18
> **Response to Reviewer 289c**
>
> Thank you so much for your insightful comments!
>
> **Q1: The negative impact of eliminating the correct label from the candidate label set.**
>
> **A:** Thanks for raising the concern. Indeed, conventional PLL methods could suffer from the negative impact of noisy PLL instances inevitably. Our countermeasure to alleviate this issue is to reduce the number of noisy PLL instances in the pruned PLL data as much as possible. For this purpose, we aim to decrease the upper bound of the pruning error by leveraging high-quality representations extracted from advanced feature extractors and using appropriate parameters in the proposed method. Empirically, the pruned PLL datasets only have a very small proportion of noisy PLL instances (as shown in Table 3), which makes PLL methods achieve impressive performance improvements. This observation definitely validates the negligible effect of noisy PLL instances caused by the proposed method. Besides, it would be also interesting to deal with potential noisy PLL instances directly during the pruning procedure.
>
> **Q2: More details about the $k$-NN algorithm and feature extractors.**
>
> **A:** Thanks for pointing out this issue. We have provided more details about the $k$-NN algorithm and the used feature extractors in Appendix C of our paper. Specifically, we employed the visual encoder of BLIP-2 [1] to extract 768-dimensional high-quality representations for all training instances in each PLL dataset. Then, we leveraged FAISS [2] to conduct the fast $k$-NN searching for each PLL instance based on the squared Euclidean Distance.
>
> **Q3: Consider the work [3] in related works and experiments.**
>
> **A:** Thanks for your suggestion! We have introduced the work [3] into the related work section of our paper. Moreover, we have conducted additional experiments by using the POP method [3] on the original and pruned PLL datasets following its suggested configurations. The experimental results on CIFAR-10 and CIFAR-100 are shown in the following tables. Note that O (vs. P) Test Acc means the test accuracy obtained by training on the original (vs. pruned) PLL dataset.
>
> |CIFAR-10|$q=0.4$|$q=0.6$|LD|ID|
> |:--: |:--: |:--: |:--: |:--: |
> |O Test Acc|95.19|94.57|95.63|93.63|
> |P Test Acc|**95.64**|**95.48**|**95.87**|**95.05**|
>
> |CIFAR-100|$q=0.05$|$q=0.1$|LD|ID|
> |:--: |:--: |:--: |:--: |:--: |
> |O Test Acc|76.35|74.38|74.90|73.36|
> |P Test Acc|**76.85**|**75.95**|**75.32**|**74.26**|
>
> From the above tables, we can see that POP equipped with our proposed CLSP method has significant performance improvements.
>
> **Q4: What is the main difference between a data-centric method and a pre-processing method?**
>
> **A:** Thanks for the interesting question. We think that our proposed data-centric method can be regarded as a special type of pre-processing method. This is because the proposed method can be used to pre-process training PLL instances before using these data to train deep PLL methods. We expect that our first data-centric method will motivate more studies on pre-processing methods for PLL.
>
> **Q5: Does the feature extractor in the $k$-NN algorithm come from the classifier during the training process? If yes, it seems unreasonable to say that the method is training-free. If not, a pre-trained model should be introduced.**
>
> **A:** Thanks for raising this concern. We would like to explain that the feature extractor does not come from the classifier during the training process. Instead, the feature extract comes from a multi-modal pre-trained model (e.g., CLIP, ALBEF, BLIP-2), which has not been trained on the corresponding PLL dataset. Hence, the proposed method is indeed training-free by leveraging a multi-modal pre-trained model. We have already introduced the pre-trained models in Section 4.1 of our original paper.
> Notably, to evaluate the effect of different types of feature extractors, we additionally trained two ResNet-18-based models (i.e., ResNet-SSL, ResNet-S) on the corresponding PLL dataset. Note that we did not use them in the proposed method and just used them for comparing the effects of different types of feature extractors. Specifically, ResNet-S was trained on each PLL dataset with the original clean supervision, while ResNet-SSL was trained on each PLL dataset by the self-supervised learning method SimCLR [4] without any supervision. More details about the training setup of these two feature extractors have been shown in Appendix C of our paper.
>
> **References:**
>
> [1] https://github.com/salesforce/LAVIS
>
> [2] https://github.com/facebookresearch/faiss
>
> [3] Progressive Purification for Instance-dependent Partial Label Learning. ICML. 2023.
>
> [4] https://github.com/google-research/simclr

---

> > ### Comment · Reviewer_289c · 2023-11-20
> >
> > The authors have adequately addressed the majority of my concerns. On another note, I would like to inquire whether the theorem and methodology presented in the paper were motivated by [1].
> >
> > [1] Zhu Z, et al. Detecting corrupted labels without training a model to predict[C]. ICML’22.

---

> > > ### Author Response · Authors · 2023-11-20
> > >
> > > Thanks for the question. We have read many data-centric related papers in various fields of machine learning, including detecting label noise [1, 2], improving data in semi-supervised learning [3], and enhancing the quantity and quality of data in long-tailed learning [4]. These studies motivated us to pioneer a data-centric study in PLL. We would like to explain that the reason for not introducing [1] in our paper was that it focused on noisy-label learning instead of PLL. Now, we have cited the paper [1] in our revised paper to further improve the clarity. Indeed, the method proposed by [1] somewhat inspired us to develop a $k$-NN-based training-free method for PLL by extracting high-quality representations with a pre-trained feature extractor. We would like to emphasize that the task of CLSP proposed in the paper is inherently different and even more challenging than detecting label noise [1, 2] because the magnitude of false candidate labels in PLL is generally larger than the number of noisy labels in noisy-label learning. As for the methodology, the proposed CLSP method in our paper depends on the down-voting of nearby instances to eliminate potential false candidate labels, while the proposed method in [1] uses the majority of votes from nearby instances to detect noisy examples. Besides, Proportion 4.1 in [1] considers the lower bound of a correct detection in the binary classification case, while Theorem 1 in our paper provides an upper bound of an incorrect pruning in the multi-class classification case. Hence, we think that the proposed method for CLSP in our paper is inherently different from the method proposed by [1] for noisy-label learning.
> > >
> > > [1] Detecting corrupted labels without training a model to predict. ICML. 2022.
> > >
> > > [2] Model-Agnostic Label Quality Scoring to Detect Real-World Label Errors. ICML. 2022.
> > >
> > > [3] A Data-Centric Approach for Improving Ambiguous Labels with Combined Semi-supervised Classification and Clustering. ECCV. 2022.
> > >
> > > [4] Data-Centric Long-Tailed Image Recognition. ArXiv:2311.01744. 2023.

---

> > > > ### Comment · Reviewer_289c · 2023-11-23
> > > >
> > > > Since all of my concerns have been addressed, I decide to raise my score to 8 (accept).

---

### Official Review · Reviewer_zBDH · 2023-10-28

**Soundness:** 4 excellent
**Presentation:** 3 good
**Contribution:** 4 excellent
**Rating:** 8
**Confidence:** 4

**Summary:**

The paper involves the partial label learning problems where each training example is equipped with multiple candidate labels instead of the only one ground-truth label provided in the conventional supervised learning setup. The paper proposes an innovative task to assistant partial label learning, i.e., candidate label set pruning, which targets at removing false candidate labels of each training example. The proposed pruning approach leverages the inconsistency of representation and label spaces to select certain candidate labels being abandoned beforehand.

The authors prove that the pruning error is upper bounded by the representation quality, the process of candidate label generation, and the pruning proportion. Moreover, they perform a numerical simulation experiment to empirically show how these factors affects the upper bound, which provides a practical guidance for selecting parameters k and $\tau$ in the proposed approach.

The paper conducts comprehensive experiments on various datasets CIFAR-10, CIFAR-100, Tiny-ImageNet, and PASCAL VOC with different settings of partial label learning including uniform -, label-dependent -, and instance-dependent candidate label generation. Besides, ten state-of-the-art partial label learning algorithms are used to compare the performance improvement. The overall experiment results show that the pruning approach enables these algorithms have a great performance gain specially on more difficult settings.

**Strengths:**

1.	Originality. The originality of the paper lies in the proposed task candidate label set pruning and the corresponding approach for the task. For what I can tell, it is the first work to propose the task for partial label learning. Moreover, the idea of leveraging the inconsistency of representation and label spaces to filter out candidate labels is also novel.

2.	Quality. The approach proposed in the paper makes both theoretical and technical contributions. The theoretical analysis of the upper bound is very interesting. The proposed approach is technically sound, which is validated by significant performance improvements in the experiment.

3.	Clarity. The paper is well organized and easy to understand the motivation. The related work is introduced adequately. The proposed task candidate label set pruning has a formal clear definition (definition 1).

4.	Significance. The paper brings a new data-centric view for the area of partial label learning, which is significant for the development of partial label learning. Perhaps, more attention of researchers could be shifted from designing complex training methods to studying efficient pruning methods.

**Weaknesses:**

1.	The numerical simulation experiment about the calculating of values and conclusions is not shown clearly enough.

2.	The PASCAL VOC dataset used in the experiment is not introduced well, as the dataset is not originally for partial label learning.

3.	Detail of trained feature extractors (ResNet-SSL, ResNet-S) is shown unclearly.

**Questions:**

1.	It would be appreciated if a clearer explanation for Definition 2 is provided. Why is this Definition needed?

2.	How are these values of k and $\gamma_{i}$ calculated in Figure 1?

3.	Why does the loss curve on VOC in Figure 5 have a rise (except PRODEN algorithm)? This phenomenon is different from other cases on CIFAR and Tiny-ImageNet datasets.

4.	How is the PASCAL VOC dataset used for partial label learning algorithms?

5.	What is the potential limitation of the proposed approach? Discussing this point is also important to have a comprehensive understanding for the proposed approach.

---

> ### Author Response · Authors · 2023-11-18
> **Response to Reviewer zBDH**
>
> Thank you so much for your insightful comments!
>
> **Q1: The numerical simulation experiment about the calculating of values and conclusions is not shown clearly enough. (How are these values of $k$ and $\gamma_{i}$ used to calculate the upper bound in Figure 1?)**
>
> **A:** Thanks for pointing out this issue. We have improved the clarity of this part in Section 3.3 of our paper. Our objective of the numerical simulation experiment is to evaluate the effects of representations of different qualities and different label ambiguities on the upper bound of the pruning error, for the proposed method. For this purpose, we empirically set various values of $\delta_{k}$ and $\rho_{k}$ where a small value of $\delta_{k}$ ($\rho_{k}$) implies high-quality representations (candidate label sets of low label ambiguity). By substituting the specific values of $\delta_{k}$ and $\rho_{k}$, we can calculate an exact upper bound with varying values of $k$ and different numbers of eliminated candidate labels $\gamma_{i}$. In this way, we are able to know how to select suitable values of $k$ and $\gamma_{i}$, with different feature extractors and PLL settings in the experiments.
>
> **Q2: The PASCAL VOC dataset used in the experiment is not introduced well, as the dataset is not originally for partial label learning. (How is it used in partial-label learning?)**
>
> **A:** Thanks for raising the concern on the used PASCAL VOC dataset. In particular, we followed the previous work [1] to construct the dataset where objects in images were cropped as instances and all objects appearing in the same original image were regarded as the labels of a candidate set. In this way, we could obtain a PLL version of the PASCAL VOC dataset and perform any PLL method on this dataset. More characteristics of PASCAL VOC have been shown in Appendix C of our paper.
>
> **Q3: The detail of trained feature extractors (ResNet-SSL, ResNet-S) is shown unclearly.**
>
> **A:** Thanks for pointing out this issue. Specifically, ResNet-S was trained on each PLL dataset with the original clean supervision using the cross-entropy loss, while ResNet-SSL was trained on each PLL dataset by the self-supervised learning method SimCLR [2] without supervision. We have further detailed the training setup of feature extractors in Appendix C of our paper.
>
> **Q4: It would be appreciated if a clearer explanation for Definition 2 is provided. Why is this Definition needed?**
>
> **A:** Particularly, Definition 2 aims to characterize the candidate label distribution in the local representation space. Specifically, a PLL instance’s $k$-NN instances are expected to have the true label in their candidate label sets with a high probability, while their candidate label sets are unlikely to have the same false candidate label. This characteristic contributes to the label distinguishability of candidate label sets in the local representation space, which is important for the proposed method to achieve a satisfying pruning error.
>
> **Q5: Why does the loss curve on VOC in Figure 5 have a rise (except the PRODEN algorithm)? This phenomenon is different from other cases on CIFAR and Tiny-ImageNet datasets.**
>
> **A:** The phenomenon is true that the training loss curves of certain PLL methods have a rise after training with pruned PLL data on PASCAL VOC in Figure 5. We reckon that there are two reasons. Firstly, there are more proportional noisy PLL instances in PASCAL VOC (the pruning error is 5.2\%) than in CIFAR and Tiny-ImageNet (the pruning error is almost < 1\%). Secondly, label disambiguation towards PLL instances in PASCAL VOC is more challenging due to the complicated visual objects in PASCAL VOC. Hence, fitting noisy PLL instances in PASCAL VOC is more difficult, thereby leading to a larger training loss value. More details are shown in Appendix D of our paper.
>
> **Q6: What is the potential limitation of the proposed approach? Discussing this point is also important to have a comprehensive understanding of the proposed approach.**
>
> **A:** Thanks for raising the concern. We think that a potential limitation of the proposed method is the presence of noisy PLL instances in the pruned PLL dataset whose true label is incorrectly eliminated and inside the non-candidate label set. Besides, many PLL methods are incapable of dealing with noisy PLL instances in the pruned PLL datasets. To alleviate this issue, as analyzed in the numerical simulation experiment, we can decrease the upper bound of the pruning error by leveraging high-quality representations extracted from advanced feature extractors and using appropriate parameters in the proposed method, thereby effectively reducing the number of noisy PLL instances in the pruned PLL dataset. In addition, it would be also interesting to deal with potential noisy PLL instances directly during the pruning procedure.
>
> **References:**
>
> [1] Long-tailed partial label learning via dynamic rebalancing. ICLR. 2023.
>
> [2] https://github.com/google-research/simclr

---

### Official Review · Reviewer_kuyu · 2023-10-31

**Soundness:** 3 good
**Presentation:** 3 good
**Contribution:** 3 good
**Rating:** 8
**Confidence:** 4

**Summary:**

The paper pioneers a data-centric study for the problem of partial-label learning (PLL) where each training instance is assigned with some additional false candidate labels along with its true label and proposes a new PLL related task named candidate label set pruning (CLSP) that aims to filter out false candidate labels of the training PLL data. To this end, the paper proposes the first kNN-based CLSP method that eliminates candidate labels of each training instance which have the high “down votes” from its kNN instances. Theoretically, the authors prove an upper bound of the pruning error and analyze the effect of representation quality and candidate generation against it. Empirically, after training with the pruned data, existing PLL algorithms have a significant performance improvement, which validates the effectiveness of the proposed method.

**Strengths:**

-	The proposed task CLSP is very significant and novel in PLL. Instead of studying learning-centric training algorithms, the authors take a different path to study filtering out false candidate labels before the training of networks, which improves the labeling quality of training PLL data and boosts the performance of existing PLL algorithms.

-	The proposed CLSP method is simple but effective, achieving impressive empirical results on various PLL settings (random, LD, ID), benchmarks (CIFAR, Tiny-ImageNet, and VOC), and ten PLL algorithms.

-	The theoretical analysis of the pruning error is very interesting. The findings in the numerical simulation experiment achieve a good guidance for the selection of parameters k and tau in the practical employment.

**Weaknesses:**

-	More empirical analysis in the experiment should be presented, such as which PLL algorithms are more sensitive to the pruning method.

-	More explanations on bad cases are needed to show the limitation of the proposed method.

**Questions:**

-	How about pruning on noisy PLL data? I am curious about whether the proposed method could be used on noisy PLL data whose true label is outside the candidate label set.

-	I find some bad cases in the experiment (shown in Table 1 and Table 8) where the performance drops after pruning. How to explain this phenomenon?

-	Could you show the values of delta_k and pho_k on the real-world dataset VOC (which are not shown in Figure 3)?

-	A unified proportion tau for each training instance is used in Eq. (2). Are there other ways to adaptively control the number of eliminated candidate labels for each training instance?

Overall, it is a good work on PLL, but there are still minor issues that can be further improved. I may consider increasing my score if the above listed weaknesses and questions can be clearly addressed.

---

> ### Author Response · Authors · 2023-11-18
> **Response to Reviewer kuyu**
>
> Thanks for your insightful comments!
>
> **Q1: More empirical analyses in the experiment should be presented.**
>
> **A:** Thanks for your suggestion! We have provided additional analyses on the experimental results of the transductive accuracy comparison in Table 7 and analyses on the training loss curves in Figure 5. For the transductive accuracy comparison in Table 7, we discover that naive PLL methods (e.g., CC, PRODEN, LWS, and CAVL) have more significant performance improvements than advanced PLL methods (e.g., PiCO, CRDPLL, ABLE, and IDGP). We think that this is because naive PLL methods have a limited capability in label disambiguation compared with advanced PLL methods, hence the proposed CLSP method has a bigger effect on promoting label disambiguation in naive PLL methods. For the training loss curves in Figure 5, we found an unusual phenomenon where the training loss values of many PLL methods become larger on PASCAL VOC after training with pruned PLL data. We reckon that there are two reasons. Firstly, there exists a higher proportion of noisy PLL instances in PASCAL VOC (where the pruning error is about 5.2\%) than in CIFAR and Tiny-ImageNet (where the pruning error is less than 1\% in most cases). Secondly, label disambiguation of training PLL instances in PASCAL VOC is more challenging due to the complicated visual objects in PASCAL VOC. Hence, fitting noisy PLL instances in PASCAL VOC is more difficult, thereby leading to a larger training loss value. More details of these empirical analyses have been shown in Appendix D of our revised paper.
>
> **Q2: More explanations on bad cases are needed to show the limitations of the proposed method. (How to explain the phenomenon of bad cases?)**
>
> **A:** Thanks for raising the concern about bad cases! We have explained the potential reason for the based cases in the experiment results of our paper. Specifically, there are a few bad cases (4/149 $\approx$2.7\%) in the experimental results where the performance of certain PLL methods has slightly degraded after training with pruned PLL data. We argue that this is because the involved PLL methods (i.e., ABLE and SoLar) in the bad cases have a time-consuming training procedure (500 and 1000 epochs respectively), hence they tend to overfit noisy PLL instances eventually, thereby leading to performance degradation.
>
> **Q3: How about pruning on noisy PLL instances?**
>
> **A:** Thank you for this interesting point! Regrettably, the proposed CLSP method may be infeasible to handle noisy PLL instances directly. This is because our method only focuses on the candidate label set and thus has no effects on noisy PLL instances whose true label is inside the non-candidate label set.  Fortunately, when the number of noisy PLL instances in the original dataset is not large, the proposed CLSP method can be still applied.
>
> **Q4: Could you show the values of $\delta_{k}$ and $\rho_{k}$ on the real-world dataset VOC?**
>
> **A:** Thanks for your suggestion. Following the empirical formulation of $\delta_{k}$ and $\rho_{k}$ shown in Section 4.1, we have conducted additional experiments to calculate their empirical values on PASCAL VOC. The calculated values of (1-$\delta_{k}$) and $\rho_{k}$ are shown in the following tables.
>
> |$k$|5|20|50|100|150|200|
> |:--: |:--: |:--: |:--: |:--: |:--: |:--: |
> |1-$\delta_{k}$|.792|.766|.736|.700|.676|.550|
>
> |$k$|5|20|50|100|150|200|
> |:--: |:--: |:--: |:--: |:--: |:--: |:--: |
> |$\rho_{k}$|.655|.604|.580|.565|.557|.550|
>
> From the above tables, we can see that the values of (1-$\delta_{k}$) and $\rho_{k}$ both decrease progressively as the value of $k$ increases. Further compared with the values on CIFAR in Figure 3, the values of (1-$\delta_{k}$) on PASCAL VOC are smaller than on CIFAR, which implies the quality of representations on PASCAL VOC is lower than on CIFAR. On the other hand, the values of $\rho_{k}$ on PASCAL VOC are larger than on CIFAR in most cases, indicating that label ambiguity in PASCAL VOC is more severe than in CIFAR. Hence, the proposed CLSP method is more challenging to employ on PASCAL VOC.
>
> **Q5: A unified proportion $\tau$ for each training instance is used in Eq. (2). Are there other ways to adaptively control the number of eliminated candidate labels for each training instance?**
>
> **A:** The answer is positive! It is feasible to eliminate different numbers of candidate labels in training PLL instances in an adaptive manner. We can formulate the problem of adaptively deciding the optimal number of eliminated candidate labels for each training PLL instance as an integer programming problem. In this case, certain optimization techniques might be required to solve the integer programming problem.

---

> > ### Comment · Reviewer_kuyu · 2023-11-23
> >
> > Thanks for your response. The authors have addressed all my concerns. I will raise my score.

---

### Meta-Review · Area_Chair_bK8h · 2023-12-05

**Metareview:**

This paper was reviewed by four experts in the field and received 8, 8, 8, 8 as the final ratings. The reviewers concurred that the proposed candidate label set pruning strategy is novel and significant in the context of partial label learning; the algorithm is theoretically sound and performs well empirically.  The reviewers raised a few questions mainly about the experimental setup, which were addressed convincingly by the authors in the rebuttal. We also appreciate the authors’ efforts in conducting additional experiments to answer the questions posed by reviewers kuyu and 289c.

The reviewers, in general, have a positive opinion about the paper and its contributions. All of them have been satisfied with the authors' rebuttal and have recommended acceptance unanimously. Based on the reviewers’ feedback, the decision is to recommend the paper for acceptance to ICLR 2024. We congratulate the authors on the acceptance of their paper!

**Justification For Why Not Higher Score:**

N/A.

**Justification For Why Not Lower Score:**

The reviewers have agreed that the paper proposes a novel method in the context of partial label learning, which is theoretically sound and works well empirically. The author rebuttal has addressed all their concerns, and all of them have recommended acceptance enthusiastically. This paper will thus be a good addition to the list of oral presentations at ICLR 2024.

---

### Decision · Program_Chairs · 2024-01-16

Accept (oral)